# Infilling Score ✳ A Pretraining Data Detection Algorithm for Large Language Models

**Negin Raoof   Litu Rout   Giannis Daras**

**Sujay Sanghavi   Constantine Caramanis   Sanjay Shakkottai   Alexandros G. Dimakis**

The University of Texas at Austin

{neginmr, litu.rout,giannisdara,constantine,sanjay.shakkottai}@utexas.edu
sanghavi@mail.utexas.edu, dimakis@austin.utexas.edu

## Abstract

In pretraining data detection, the goal is to detect whether a given sentence is in the dataset used for training a Large Language Model (LLM). Recent methods (such as Min-K% and Min-K%++) reveal that most training corpora are likely contaminated with both sensitive content and evaluation benchmarks, leading to inflated test set performance. These methods sometimes fail to detect samples from the pretraining data, primarily because they depend on statistics composed of causal token likelihoods. We introduce Infilling Score, a new test-statistic based on non-causal token likelihoods. Infilling Score can be computed for autoregressive models without re-training using Bayes rule. A naive application of Bayes rule scales linearly with the vocabulary size. However, we propose a ratio test-statistic whose computation is invariant to vocabulary size. Empirically, our method achieves a significant accuracy gain over state-of-the-art methods including Min-K%, and Min-K%++ on the WikiMIA benchmark across seven models with different parameter sizes. Further, we achieve higher AUC compared to reference-free methods on the challenging MIMIR benchmark. Finally, we create a benchmark dataset consisting of recent data sources published after the release of Llama-3; this benchmark provides a statistical baseline to indicate potential corpora used for Llama-3 training.

## 1 Introduction

The significant progress in language modeling can largely be attributed to development and deployment of large-scale models that utilize extensive training corpora, often encompassing trillions of tokens (Li et al., 2024; Dubey et al., 2024). The selection and curation of data for training such Large Language Models (LLMs) is very complex and expensive. Further, recent developers of LLMs withhold details regarding the sources of their pretraining datasets (Dubey et al., 2024; OpenAI et al., 2024; Touvron et al., 2023b). This lack of transparency has raised concerns regarding the inadvertent inclusion of copyrighted content (Chang et al., 2023; Min et al., 2023; Meeus et al., 2023) or personally identifiable information (Mozes et al., 2023; Panda et al., 2024), potentially leading to ethical and legal challenges (Grynbaum & Mac, 2023). Furthermore, the inclusion of benchmark datasets within the training corpora itself can compromise the integrity of model evaluations. This practice may inflate test performance metrics without accurately reflecting the model's capabilities (Oren et al., 2023; Zhou et al., 2023).

Recent work has focused on the problem of determining whether specific sequences of tokens have been previously seen by a language model (Shi et al., 2024; Zhang et al., 2024; Duan et al., 2024). These investigations are categorized under a growing field of attacks on LLMs known as Membership Inference Attacks (MIA) (Shokri et al., 2017; Mattern et al., 2023b; Carlini et al., 2022). Many studies in this area focus on fine-tuning data detection (Song & Shmatikov, 2019; Shejwalkar et al., 2021; Mahloujifar et al., 2021). However, pretraining data detection attacks are becoming increasingly important as they can reveal whether a model has been trained on potentially sensitive data and prevent evaluation data contamination (Jiang et al., 2024; Yang et al., 2023).

We introduce a novel method for identifying whether a given text sequence was part of a language model's pretraining data. Our method uses a new test-statistic that we call the *Infilling Score*. Our approach performs a non-causal test to compute the infilling probability of a token, based on the tokens that appear *before and after* this token in the sentence. An autoregressive language model generates causal likelihoods (i.e. the probability of a word appearing *after* some context). We find that non-causal likelihoods lead to more accurate tests for membership inference. These likelihoods can be computed using a causal autoregressive model. The computation involves applying Bayes' rule and the law of total probability, and needs a marginalization over the vocabulary to compute a partition function. Unfortunately, computing this partition function requires calling the autoregressive LM many times, one for each vocabulary entry. This would require tens of thousands of calls to the autoregressive LLM to compute a single non-causal probability for one token, and hence is not practical. Our central idea is to propose an approximate test-statistic whose computation is much faster, does not require an exact computation of this partition function and does not depend on the vocabulary size.

Our method achieves a significant accuracy gain over state-of-the-art methods including Min-K%, and Min-K%++ on the WikiMIA benchmark across seven models. On WikiMIA, our method outperforms the previous state of the art in AUC. It achieves up to $10\%$ improvement on Llama models when testing long sequences (256 tokens). Further, we achieve higher AUC compared to reference-free methods on the challenging MIMIR benchmark.

**Our main contributions are summarized below:**

(1) We introduce the Infilling Score, a new reference-free method for detecting pretraining data using infilling likelihood of tokens within the candidate sentence (Section 3). While SoTA methods: MIN-K% and MIN-K%++ rely on a statistic based on past tokens only, our method computes a new test statistic considering both past and future tokens in the sentence.

(2) We develop an efficient algorithm for computing this new score. Though our method conceptually shares similarities with a likelihood computed via Bayes rule, computationally it is much different: whereas any natural approach for computing a Bayes rule calculation scales with vocabulary size, our algorithm has computation invariant to vocabulary size.

(3) We conduct extensive experiments on the standard (a) WikiMIA (Shi et al., 2024) and, (b) MIMIR (Duan et al., 2024) to verify the efficacy of our method (Section 4). On these benchmarks, we compare our method with state-of-the-art MIA methods including MIN-K% (Shi et al., 2024) and MIN-K%++ (Zhang et al., 2024). On WikiMIA, our method achieves 11% improvement over MIN-K% and 5% improvement over MIN-K%++ in terms of AUROC on average. We attribute the notable performance gain of our method to infilling probability (Section 3).

(4) We curate a dataset of book excerpts that have not been seen by the LLMs released before April 2024 (Section 4.1). Employing our Infilling Score, we detect a list of books which have (likely) been used for training Llama-3-8B (Dubey et al., 2024) (4.4.3).

## 2 BACKGROUND

In this section, we discuss the standard definition of Membership Inference Attack (MIA) and recent advances along this line of research.

**Problem setup.** Given a sentence $\boldsymbol{x} = \{x_i\}_{i=0}^N$ and a Large Language Model (LLM) denoted by $\mathcal{M}$, the goal of MIA is to build a detector $h(\boldsymbol{x}, \mathcal{M}) \rightarrow \{0, 1\}$ that can infer the membership of $\boldsymbol{x}$ in the training corpus $\mathcal{D} = \{\boldsymbol{x}_j\}_{j \in [n]}$ of $\mathcal{M}$. Existing MIA methods for LLMs (Shi et al., 2024; Zhang et al., 2024; Carlini et al., 2021; Mattern et al., 2023a) assign a score to each sample $\boldsymbol{x}$ and use a binary threshold to determine its membership class, with 1 indicating $\boldsymbol{x} \in \mathcal{D}$ and 0 otherwise.

### 2.1 CHALLENGES IN PRETRAINING DATA DETECTION USING MIA METHODS

#### 2.1.1 DETECTION DIFFICULTY

Prior works (Hardt et al., 2016; Bassily et al., 2020) have shown that the total variation (TV) distance between the distribution of seen and unseen data is proportional to the learning rate, size of the

dataset $|\mathcal{D}|$ and the frequency of the test sentence $x$. Since TV captures the separability between these distributions, low TV makes it difficult to infer the membership class of a given $x$.

### 2.1.2 ARCHITECTURE AND PRETRAINING DISTRIBUTION

Membership inference attacks for LLM pretraining data detection are broadly categorized into two classes: (a) reference-based methods and (b) reference-free methods. **Reference-based methods** such as Reference (Carlini et al., 2021) infer the membership of a sentence $x$ by computing the likelihood of $x$ using two different LLMs. They compare the perplexity of $x$ under the target LLM with the perplexity of $x$ under a smaller language model. The smaller model $\mathbb{M}$ shares the same architecture as $\mathcal{M}$, and is trained on a subset of samples, $\mathbb{D}$, collected from the same underlying distribution of $\mathcal{D}$. The intuition is that smaller networks have less capacity to memorize sentences from the pretraining dataset. One crucial limitation of these methods is that reference model may not always exist. Although LLM developers often do not disclose information about the distribution of pretraining data, reference-based MIAs (Carlini et al., 2021) assume the knowledge of the architecture and underlying pretraining distribution, making these methods less practical.

Among **reference-free methods**, Min-K% (Shi et al., 2024) hypothesizes that when a sentence is seen by the model, i.e., $x \in \mathcal{D}$, it usually contains a number of tokens with low causal probabilities (outliers). Formally, given a sequence of tokens $x = \{x_i\}_{i=0}^N$, Min-K% score is given by:

$$\text{Min-K\%}(\boldsymbol{x}) = \frac{1}{|\text{min-}k\%|} \sum_{x_i \in \text{min-}k\%} \text{Min-K\%}_{\text{token}}(x_i), \tag{1}$$

$$\text{where Min-K\%}_{\text{token}}(x_i) = \log p\left(x_i | x_{<i}\right). \tag{2}$$

Here, $\text{Min-K\%}_{\text{token}}(x_i)$ denotes the score for each token $x_i$. The set min-$k\%$ contains $k\%$ of the input tokens which correspond to the bottom $k\%$ scores within the sequence. If the average score for this set is less than $\tau(k)$, where $\tau(k)$ denotes the binary threshold for a fixed $k$, then Min-K% detects the sequence $x$ as "unseen". Note that the classification threshold $\tau(k)$ is determined empirically using a validation dataset.

A recently proposed method, **Min-K%++** (Zhang et al., 2024), improves the detection accuracy of Min-K% by normalizing the next-tokens log likelihood $\log p(x_i|x_{<i})$ as follows:

$$\text{Min-K\%++}(\boldsymbol{x}) = \frac{1}{|\text{min-}k\%|} \sum_{x_i \in \text{min-}k\%} \text{Min-K\%++}_{\text{token}}(x_i), \tag{3}$$

$$\text{where Min-K\%++}_{\text{token}}(x_i) = \frac{\log p\left(x_i | x_{<i}\right) - \mu_{x_{<i}}}{\sigma_{x_{<i}}} \tag{4}$$

$\mu_{x_{<i}} = \mathbb{E}_{z \sim p(.|x_{<i})}[\log p(z|x_{<i})]$ and $\sigma_{x_{<i}} = \sqrt{\mathbb{E}_{z \sim p(.|x_{<i})}[(\log p(z|x_{<i}) - \mu_{x_{<i}})^2]}$ are the mean and standard deviation of the next-token likelihood.

Both Min-K% and Min-K%++ rely on the "causal" likelihood predictions of the model. However, the causal likelihood of $x_i$ does not consider the information from the entire sentence context, as it only depends on the preceding tokens $x_{<i}$. We propose that sentences seen during training ($\boldsymbol{x} \in \mathcal{D}$) typically have a number of tokens with low **infilling** probabilities. By using the non-causal token likelihoods which depend on both preceding and succeeding tokens ($x_{<i}$ and $x_{>i}$), we achieve a more accurate statistic than causal likelihoods alone. This enables our Infilling Score method to outperform previous pretraining data detection approaches on standard benchmarks.

## 3 METHOD

We describe our method in this section. First we describe the computation of our new ratio statistic, and explain why it offers computational scalability compared to a straightforward application of Bayes Rule. Next, we describe how this score is used to detect data in the pretraining set. Finally, we explain how we employ our method to detect pretraining samples in Llama-3.

| Ground truth: | She | ate | Italian | pasta |
|---|---|---|---|---|
| Masked input: | She | ate | <MASKED> | pasta |
| | $x_1$ | $x_2$ | $m_3$ | $x_4$ |

## 3.1 Computing the Infilling Likelihood

In this setting, we search for the most likely token to infill $m_3$ using other tokens in the sentence, i.e., $\{x_1, x_2, x_4\}$. Using the law of total probability, we get:

$$p(x_3|x_1, x_2, x_4) = \frac{p(x_4|x_1, x_2, x_3)p(x_3|x_1, x_2)}{p(x_4|x_1, x_2)} = \frac{p(x_4|x_1, x_2, x_3)p(x_3|x_1, x_2)}{\sum_{x_3' \in \mathcal{V}} p(x_4|x_1, x_2, x_3')p(x_3'|x_1, x_2)}. \quad (5)$$

Observe that the partition function in the denominator of equation 5 is expensive to compute as it requires summation over all the tokens in the vocabulary $\mathcal{V}$. In the naive case, *the number of LLM calls required to compute the infilling likelihood scales linearly both with vocabulary size and the sequence length.* This is because for each token, the denominator in equation 5 scales linearly in the vocabulary size, and this computation needs to be repeated for each token. The size of the vocabulary can be as large as $128K$ in recent LLM (Dubey et al., 2024).

To address the scalability challenge, we introduce a ratio test statistic. Our main idea is to compute the ratio of the infilling probability of the ground-truth token and the maximum causal likelihood token. Using this proposed statistic, we bypass the need to compute the computationally expensive partition function. In the above setting, we define the ratio test-statistic of token $x_3$ as:

$$\frac{p(x_3|x_1, x_2, x_4)}{p(x_3^*|x_1, x_2, x_4)}, \text{ where } x_3^* = \underset{x_3' \in \mathcal{V}}{\arg\max}\, p(x_3'|x_1, x_2). \quad (6)$$

This ratio compares the infilling likelihood of the ground-truth token to that of the model's causal prediction. If $x_3$ is an outlier the ratio is closer to 0, and when $x_3$ is among the model's top predictions, this ratio is closer to 1. Since the partition function in equation 5 is the same for $p(x_3|x_1, x_2, x_4)$ and $p(x_3^*|x_1, x_2, x_4)$, it is canceled in the ratio test statistic. This drastically reduces the number of LLM calls from $\mathcal{O}(N|\mathcal{V}|)$ to $\mathcal{O}(N)$, making our test statistic independent of the size of the vocabulary (details in 4.5). Interestingly, we can exactly compute this ratio analytically using auto-regressive models without retraining. We then compute the log of this ratio and normalize the probabilities to capture the relative significance of each token in the vocabulary. First, we derive

$$\log \frac{p(x_3|x_1, x_2, x_4)}{p(x_3^*|x_1, x_2, x_4)} = \log \frac{p(x_4|x_1, x_2, x_3)p(x_3|x_1, x_2)}{p(x_4|x_1, x_2, x_3^*)p(x_3^*|x_1, x_2)} \quad (7)$$

$$= \log p(x_4|x_1, x_2, x_3) + \log p(x_3|x_1, x_2) - \log p(x_4|x_1, x_2, x_3^*) - \log p(x_3^*|x_1, x_2), \quad (8)$$

Generalizing (equation 7) to use $m$ future tokens for calculating the infilling ratio of token $i$, we get the following:

$$\log \frac{p(x_i|x_{1:i-1}, x_{i+1:n})}{p(x_i^*|x_{1:i-1}, x_{i+1:i+m})} =$$

$$\sum_{j=i+1}^{i+m} \log p(x_j|x_1, x_2, ..., x_i, ...x_{j-1}) + \log p(x_i|x_{1:i-1}) -$$

$$\sum_{j=i+1}^{i+m} \log p(x_j|x_1, x_2, ..., x_i^*, ...x_{j-1}) - \log p(x_i^*|x_{1:i-1}), \quad (9)$$

where $x_{1:i}$ denotes the sequence $x_1, x_2, ...x_i$, and $x_i^* = \arg\max_{x_i' \in \mathcal{V}} p(x_i'|x_{1:i-1})$.

As suggested in Zhang et al. (2024), we normalize the terms to compute our infilling score for a given token $x_3$:

$$
\begin{aligned}
\text{InfillingScore}_{\text{token}}(x_i) = & \sum_{j=i+1}^{i+m} \frac{\log p(x_j | x_1, x_2, ..., x_i, ...x_{j-1}) - \mu_{x_{1:j}}}{\sigma_{x_{1:j}}} + \frac{\log p(x_i | x_{1:i-1}) - \mu_{x_{1:i}}}{\sigma_{x_{1:i}}} \\
& - \sum_{j=i+1}^{i+m} \frac{\log p(x_j | x_1, x_2, ..., x_i^*, ...x_{j-1}) - \mu_{x_{1:j}}}{\sigma_{x_{1:j}}} - \frac{\log p(x_i^* | x_{1:i-1}) - \mu_{x_{1:i}}}{\sigma_{x_{1:i}}}
\end{aligned}
$$

$$(10)$$

where $\mu_{x_{1:j}} = \mathbb{E}_{z \sim p(.|x_{1:j})}[\log p(z | x_{1:j})]$, and $\sigma_{x_{1:j}} = \sqrt{\mathbb{E}_{z \sim p(.|x_{1:j})}[(\log p(z | x_{1:j}) - \mu_{x_{1:j}})^2]}$, are the mean and standard deviation of the next token log probability, $\log p(x_j | x_1, ..., x_{j-1})$, over the whole vocabulary. In contrast to equation 5, there is no normalization in the denominator needed in equation 10. Note that the non-causal terms in equation 6 are all replaced by causal terms which can be computed through LLM logits. To implement, we need two calls to the LLM – the first with input as the sequence $x_1, ..., x_i, ..., x_N$ and the second call with input as $x_1, ..., x_i^*, ..., x_N$. Note that the means and standard deviations can be computed from these logits. Thus, equation 10 requires only two calls to the LLM per token. Hence with $N$ tokens, the total number of calls to the LLM scales as $2N$, in contrast to the naive approach where the scaling is $N|\mathcal{V}|$. We will see in our experiments (see Section 4.5) that this leads to a dramatic decrease in runtime, with two orders of magnitude improvement.

## 3.2 PRETRAINING DATA DETECTION

To detect the membership of a given sentence $x$, we find the set of min-$k\%$ tokens with low Infilling Scores in the sentence, and compute the average score over this subset. Our final test-statistic becomes:

$$
\text{InfillingScore}(x) = \frac{1}{|\text{min-}k\%|} \sum_{x_i \in \text{min-}k\%} \text{InfillingScore}_{\text{token}}(x_i). \tag{11}
$$

Our experiments suggest that InfillingScore($x$) is higher for a given sentence $x$ which was seen by the model during pretraining. Thus, the infilling score enables us to build a detector $h(\cdot, \mathcal{M})$ for an LLM $\mathcal{M}$ to infer the membership class of $x$ as:

$$
h(x, \mathcal{M}) = \begin{cases} 0 & \text{InfillingScore}(x) < \tau \\ 1 & \text{otherwise} \end{cases}, \tag{12}
$$

where $\tau$ denotes the binary threshold that is applied on the soft scores.

## 4 EXPERIMENTS

### 4.1 BENCHMARKS

We conduct comprehensive tests to evaluate the performance of our newly proposed test-statistic against state-of-the-art reference-based and reference-free methods. We experiment with various models and different parameter sizes. Initially, we examine the established pretraining data detection benchmarks: WikiMIA (Shi et al., 2024) and MIMIR (Duan et al., 2024). WikiMIA is a temporal MIA dataset commonly used for evaluating pretraining data detection methods. This benchmark contains excerpts from Wikipedia event articles, and classifies samples based on the timestamp of the articles. Samples from articles published before the training of an LLM are classified as "seen", and samples after the training are classified as "unseen". Hence, this benchmark applies only to a subset of LLMs, depending on their training and release time. WikiMIA has four different subsets with sequence lengths of 32, 64, 128, and 256. Zhang et al. (2024) also published a "Paraphrased" version of WikiMIA which uses ChatGPT to paraphrase the samples.

A more challenging benchmark, MIMIR (Duan et al., 2024), aims to evaluate pretraining data detection methods when the distributions of "seen" and "unseen" text samples have high n-gram overlap. MIMIR consists of samples from the Pile (Gao et al., 2020) across seven domains: English

Wikipedia, ArXiv, Github, Pile CC, PubMed Central, DM Mathematics, and HackerNews. Parts from the train subset of the Pile are labeled as "seen" while parts of the test set are labeled as "unseen". These seen and unseen samples are selected to have very high n-gram overlaps, making it significantly more challenging to infer training data membership.

Previous membership inference benchmarks such as WikiMIA, BookMIA (Shi et al., 2024), and BookTection (Duarte et al., 2024) cannot be reliably used for Llama-3 because the model was trained more recently. To address this, we curate a new dataset consisting of book excerpts published after the release of Llama-3 labeled as "unseen" data. In this new dataset the "seen" data comes from classical fiction books published before 1965. We sample a set of 100 excerpts, with each excerpt containing 200 tokens. The "unseen" data consists of excerpts from books published after April 2024, similarly having size of 200 tokens.

## 4.2    Models and Metrics

We use the WikiMIA benchmark to evaluate our Infilling Score method on Llama (7B, 13B, 30B) (Touvron et al., 2023a), Pythia (2.8B, 6.9B) (Biderman et al., 2023), GPT-NeoX-20B (Black et al., 2022), and Mamba-1.4B (Gu & Dao, 2023) models. WikiMIA is applicable to models released between 2017 and 2023. Samples from the Wikipedia event articles published in and after 2023 are labeled as "unseen", and samples from articles published before 2017 are labeled as "unseen".

For experiments on the MIMIR benchmark, we evaluate our method using Pythia (160M and 1.4B) on a subset of the Pile (Gao et al., 2020) dataset sampled across seven different domains. Pythia model has been pretrained on the training set of the Pile dataset (Biderman et al., 2023). Therefore, MIMIR benchmark has labeled samples from the train/test of the Pile as "seen"/"unseen", respectively.

We evaluate Infilling Score for membership classification against the state-of-the-art methods using the area under the ROC curve (AUROC) metric. As suggested in prior studies (Carlini et al., 2022; Mireshghallah et al., 2022), we also report the True Positive rate at low False Positive rate (TPR@5%FPR).

## 4.3    Baselines

We compare our proposed method with multiple state-of-the-art methods as our baselines. Reference method (Carlini et al., 2021) relies on the ratio of the sample perplexity (e.g. next token likelihood) estimated by the target model to the sample perplexity estimated by a smaller reference model. Zlib is another reference-based method which uses the Zlib compression entropy for calibrating the score (Carlini et al., 2021). Neighbor method (Mattern et al., 2023a) replaces tokens within a sequence using a pretrained masked language model to generate similar sentences. The method identifies if a sample belongs to the training data by comparing the loss of the original sample with the average loss of its neighboring sentences. The same algorithm is also used for detecting machine generated text in (Mitchell et al., 2023). We compare our results with both Min-K%(Shi et al., 2024) and Min-K%++ methods (Zhang et al., 2024) extensively for performance evaluations because both methods are the current state-of-the-art reference-free baselines, falling under the same category as our Infilling Score.

## 4.4    Results

### 4.4.1    Evaluation on WikiMIA

Table 1 presents the results comparing our Infilling Score method with state-of-the-art methods evaluated on the WikiMIA benchmark. In addition, we evaluate the effectiveness of our method using TPR at low FPR in Table 2. Our experimental setup is consistent with prior work such as Min-K%++ and Min-K%. For 32-token sequences we only use one future token, and for longer sequences we use 5 future tokens. We fix $k = 20\%$ across all experiments.

On average, our method shows a 5% improvement in AUC over Min-K%++ across various model sizes and different inputs sequence lengths. As hypothesized in Section 3, Infilling Score consistently outperforms existing reference-based and reference-free methods in detecting Llama pretraining data. We empirically show that predicting the token-level likelihoods, using the information in both the past and future tokens is more accurate for pretraining data detection. For longer sequences. This is specially helpful for samples with longer sequence lengths where there are more tokens in the context

| Seq. length | Method | Mamba-1.4B | | NeoX-20B | | Pythia-2.8B | | Pythia-6.9B | | Llama-7B | | Llama-13B | | Llama-30B | | Average |
|---|---|---|---|---|---|---|---|---|---|---|---|---|---|---|---|---|
| | | Ori. | Para. | Ori. | Para. | Ori. | Para. | Ori. | Para. | Ori. | Para. | Ori. | Para. | Ori. | Para. | |
| 32 | Infilling Score (Ours) | 66.6 | 66.1 | 75.6 | 73.1 | 65.0 | 63.9 | 69.7 | 68.2 | 88.1 | 88.0 | 88.6 | 87.0 | 87.3 | 84.7 | 76.56 |
| | Min-K%++ (Zhang et al., 2024) | 66.8 | 66.1 | 75.0 | 69.6 | 64.4 | 62.4 | 70.3 | 68.0 | 85.1 | 84.0 | 84.8 | 82.7 | 84.3 | 81.2 | 74.62 |
| | Min-K%(Shi et al., 2024) | 63.2 | 62.9 | 71.8 | 69.7 | 61.8 | 61.7 | 66.3 | 65.2 | 68.0 | 67.0 | 68.0 | 68.4 | 70.1 | 70.7 | 66.65 |
| | Neighbor (Mattern et al., 2023a) | 64.1 | 63.6 | 70.2 | 68.3 | 62.1 | 64.5 | 65.8 | 65.5 | - | - | 65.8 | 65.0 | 67.6 | 66.3 | 65.73 |
| | Zlib (Carlini et al., 2021) | 61.9 | 62.3 | 69.0 | 68.2 | 62.1 | 62.3 | 64.3 | 64.2 | 66.7 | 67.3 | 67.8 | 68.3 | 69.8 | 70.4 | 66.04 |
| | Ref (Carlini et al., 2021) | 62.2 | 62.3 | 67.2 | 66.3 | 61.3 | 61.2 | 63.6 | 63.5 | - | - | 57.9 | 56.2 | 63.5 | 62.4 | 62.3 |
| 64 | Infilling Score (Ours) | 67.3 | 62.9 | 76.8 | 73.1 | 65.7 | 58.9 | 71.4 | 64.2 | 89.7 | 86.8 | 90.1 | 84.5 | 88.3 | 81.2 | 75.78 |
| | Min-K%++ (Zhang et al., 2024) | 67.2 | 63.3 | 76.0 | 67.5 | 65.0 | 58.5 | 71.6 | 64.8 | 85.7 | 80.8 | 86.7 | 78.8 | 84.7 | 74.9 | 73.25 |
| | Min-K%(Shi et al., 2024) | 62.2 | 58.0 | 72.2 | 66.1 | 61.2 | 56.8 | 65.0 | 61.1 | 63.3 | 61.8 | 66.0 | 64.0 | 68.5 | 65.7 | 63.71 |
| | Neighbor (Mattern et al., 2023a) | 60.6 | 60.6 | 67.1 | 67.4 | 61.3 | 59.6 | 63.2 | 63.1 | - | - | 64.1 | 64.7 | 67.1 | 66.7 | 63.79 |
| | Zlib (Carlini et al., 2021) | 60.4 | 59.1 | 67.6 | 66.4 | 60.5 | 59.0 | 62.6 | 61.6 | 63.4 | 63.6 | 65.3 | 65.3 | 67.5 | 67.4 | 63.55 |
| | Ref (Carlini et al., 2021) | 60.6 | 59.6 | 65.7 | 65.9 | 59.6 | 59.2 | 62.4 | 62.9 | - | - | 63.4 | 60.9 | 69.0 | 65.4 | 63.88 |
| 128 | Infilling Score (Ours) | 69.6 | 66.6 | 78.1 | 74.9 | 67.1 | 64.1 | 70.4 | 67.5 | 87.6 | 83.4 | 88.3 | 83.5 | 86.7 | 79.5 | 76.23 |
| | Min-K%++ (Zhang et al., 2024) | 68.8 | 65.6 | 75.9 | 72.2 | 66.8 | 63.4 | 70.4 | 66.8 | 85.7 | 82.2 | 83.9 | 76.3 | 82.6 | 73.8 | 73.88 |
| | Min-K%(Shi et al., 2024) | 66.8 | 64.5 | 75.0 | 72.6 | 66.9 | 64.7 | 69.5 | 67.0 | 70.1 | 68.1 | 71.5 | 68.7 | 73.9 | 70.2 | 69.25 |
| | Neighbor (Mattern et al., 2023a) | 64.8 | 62.6 | 71.6 | 69.6 | 65.2 | 61.9 | 67.5 | 64.3 | - | - | 68.3 | 64.0 | 72.2 | 67.2 | 66.60 |
| | Zlib (Carlini et al., 2021) | 65.6 | 65.3 | 71.8 | 71.8 | 65.0 | 65.0 | 67.6 | 67.4 | 68.3 | 68.4 | 69.7 | 69.6 | 71.8 | 71.5 | 68.48 |
| | Ref (Carlini et al., 2021) | 65.2 | 61.1 | 67.8 | 67.8 | 59.6 | 59.5 | 63.3 | 62.9 | - | - | 62.6 | 59.7 | 71.9 | 70.0 | 64.28 |
| 256 | Infilling Score (Ours) | 70.1 | - | 77.0 | - | 73.6 | - | 70.5 | - | 96.6 | - | 95.3 | - | 89.8 | - | 81.84 |
| | Min-K%++ (Zhang et al., 2024) | 65.5 | - | 71.9 | - | 63.9 | - | 65.5 | - | 82.5 | - | 82.3 | - | 77.3 | - | 72.70 |
| | Min-K%(Shi et al., 2024) | 69.8 | - | 78.0 | - | 70.0 | - | 71.1 | - | 72.4 | - | 72.9 | - | 72.1 | - | 72.33 |
| | Zlib (Carlini et al., 2021) | 67.6 | - | 73.2 | - | 69.3 | - | 69.8 | - | 71.2 | - | 73.1 | - | 72.8 | - | 71.00 |

Table 1: AUROC results on the Original and Paraphrased subsets of the WikiMIA benchmark (Shi et al., 2024). Note that the paraphrased version of the 256-token subset of WikiMIA is not published on HuggingFace which is why some results are missing for 256 tokens. Bold shows the best result and underline shows the second best results in each section. As seen, our Infilling Score method outperforms previous work for detecting pretraining samples for EleutherAI's Pythia (Biderman et al., 2023) and GPT-NeoX (Black et al., 2022), Mamba (Gu & Dao, 2023), and Meta's Llama (Touvron et al., 2023a) models across various model sizes.

| Seq. length | Method | Mamba-1.4B | | NeoX-20B | | Pythia-2.8B | | Pythia-6.9B | | Llama-7B | | Llama-13B | | Llama-30B | | Average |
|---|---|---|---|---|---|---|---|---|---|---|---|---|---|---|---|---|
| | | Ori. | Para. | Ori. | Para. | Ori. | Para. | Ori. | Para. | Ori. | Para. | Ori. | Para. | Ori. | Para. | |
| 32 | Infilling Score (Ours) | 14.0 | 16.5 | 27.6 | 23.0 | 13.7 | 13.7 | 17.3 | 20.7 | 34.1 | 35.9 | 30.5 | 29.7 | 33.1 | 38.2 | 24.85 |
| | Min-K%++ (Zhang et al., 2024) | 12.9 | 10.6 | 19.4 | 12.9 | 14.2 | 13.9 | 17.1 | 17.1 | 33.6 | 31.5 | 38.5 | 35.9 | 31.3 | 27.4 | 22.59 |
| | Min-K%(Shi et al., 2024) | 14.7 | 15.2 | 27.9 | 19.6 | 17.1 | 16.5 | 17.8 | 21.7 | 15.2 | 16.0 | 18.9 | 17.6 | 21.2 | 18.1 | 18.39 |
| | Neighbor (Mattern et al., 2023a) | 11.9 | 7.2 | 22.2 | 15.2 | 15.0 | 8.5 | 16.5 | 9.6 | - | - | 11.6 | 8.5 | 9.3 | 9.3 | 12.07 |
| | Zlib (Carlini et al., 2021) | 15.5 | 13.2 | 19.9 | 18.6 | 15.8 | 14.5 | 16.3 | 12.7 | 13.7 | 14.2 | 11.6 | 15.0 | 14.5 | 15.0 | 15.03 |
| | Ref (Carlini et al., 2021) | 7.8 | 5.9 | 1.5 | 15.2 | 6.2 | 7.2 | 6.7 | 6.2 | - | - | 4.7 | 5.4 | 9.8 | 7.5 | 7.01 |
| 64 | Infilling Score (Ours) | 19.4 | 10.2 | 27.8 | 21.8 | 18.0 | 13.4 | 21.1 | 14.8 | 50.7 | 28.5 | 53.5 | 34.9 | 44.0 | 27.8 | 27.56 |
| | Min-K%++ (Zhang et al., 2024) | 16.6 | 7.0 | 20.4 | 13.0 | 16.2 | 9.9 | 26.1 | 14.1 | 39.4 | 26.8 | 34.1 | 26.4 | 36.3 | 21.5 | 21.98 |
| | Min-K%(Shi et al., 2024) | 19.4 | 8.4 | 20.4 | 17.6 | 18.3 | 11.3 | 19.0 | 12.7 | 14.4 | 13.7 | 17.2 | 13.4 | 17.6 | 14.4 | 15.55 |
| | Neighbor (Mattern et al., 2023a) | 8.8 | 9.5 | 13.0 | 18.3 | 10.2 | 11.3 | 10.9 | 12.7 | - | - | 10.2 | 14.4 | 9.9 | 11.6 | 11.73 |
| | Zlib (Carlini et al., 2021) | 14.1 | 15.1 | 16.6 | 19.4 | 14.4 | 16.6 | 16.2 | 15.8 | 11.3 | 14.8 | 12.7 | 13.4 | 15.5 | 16.9 | 15.20 |
| | Ref (Carlini et al., 2021) | 4.6 | 8.1 | 15.5 | 14.1 | 10.6 | 13.0 | 12.0 | 16.2 | - | - | 4.2 | 4.6 | 11.3 | 8.1 | 10.19 |
| 128 | Infilling Score (Ours) | 16.6 | 15.8 | 25.9 | 33.1 | 15.8 | 13.4 | 20.9 | 21.6 | 38.1 | 33.8 | 41.0 | 30.9 | 24.5 | 31.7 | 25.93 |
| | Min-K%++ (Zhang et al., 2024) | 16.6 | 10.1 | 23.0 | 19.4 | 17.3 | 14.4 | 22.3 | 21.6 | 46.8 | 38.8 | 41.0 | 21.5 | 38.1 | 21.6 | 25.18 |
| | Min-K%(Shi et al., 2024) | 16.6 | 14.4 | 25.2 | 22.3 | 13.7 | 14.4 | 18.0 | 17.3 | 19.4 | 21.6 | 25.9 | 14.4 | 23.7 | 18.7 | 18.97 |
| | Neighbor (Mattern et al., 2023a) | 15.8 | 13.7 | 15.8 | 18.7 | 8.6 | 12.2 | 10.8 | 17.3 | - | - | 12.9 | 11.6 | 15.1 | 14.4 | 13.91 |
| | Zlib (Carlini et al., 2021) | 19.4 | 17.3 | 23.0 | 21.6 | 18.7 | 16.6 | 20.9 | 20.9 | 14.4 | 18.7 | 18.7 | 16.9 | 18.0 | 19.4 | 18.89 |
| | Ref (Carlini et al., 2021) | 10.1 | 11.5 | 15.8 | 19.4 | 10.1 | 7.2 | 13.7 | 8.6 | - | - | 10.8 | 8.1 | 10.8 | 18.7 | 12.07 |
| 256 | Infilling Score (Ours) | 25.5 | - | 29.4 | - | 19.6 | - | 29.4 | - | 80.4 | - | 80.4 | - | 72.5 | - | 48.17 |
| | Min-K%++ (Zhang et al., 2024) | 15.7 | - | 13.7 | - | 13.7 | - | 11.8 | - | 47.1 | - | 37.3 | - | 19.6 | - | 22.70 |
| | Min-K%(Shi et al., 2024) | 13.7 | - | 21.6 | - | 13.7 | - | 15.7 | - | 17.6 | - | 19.6 | - | 13.7 | - | 16.51 |
| | Zlib (Carlini et al., 2021) | 23.5 | - | 23.5 | - | 19.6 | - | 27.5 | - | 21.6 | - | 27.5 | - | 29.4 | - | 24.66 |

Table 2: True Positive rate at low False Positive rate (FPR=5%) results on the Original and Paraphrased subsets of the WikiMIA benchmark (Shi et al., 2024). Note that the paraphrased version of the 256-token subset of WikiMIA is not published on HuggingFace, which is why some results are missing for 256 tokens. Bold shows the best results and underline shows the second best results in each section. As shown, our Infilling Score method on average achieves higher True Positive rate compared to existing methods, with the best performance on 256-token long sequences.

to use for inference. Since our method offers the capability to leverage the future as well as past tokens, it shows a significant gain over current state-of-the-art method when input sequences are long.

### 4.4.2 EVALUATION ON MIMIR

Table 3 shows the results comparing our Infilling Score method with SoTA methods evaluated on the challenging MIMIR benchmark. In the MIMIR dataset, samples from the "seen" and "unseen" classes are sampled from the same dataset to ensure 13-gram overlap of up to 0.8 between the classes. Reference-based models show high performance on this benchmark. However, the drawback of this

| Method | Wikipedia 160M | Wikipedia 1.4B | Github 160M | Github 1.4B | Pile CC 160M | Pile CC 1.4B | PubMed Central 160M | PubMed Central 1.4B |
|---|---|---|---|---|---|---|---|---|
| Infilling Score (Ours) | 49.7 | 53.4 | 65.5 | 70.0 | **53.7** | **53.3** | 52.3 | **53.5** |
| Min-K%++ (Zhang et al., 2024) | 49.7 | 53.7 | 64.8 | 69.6 | 50.6 | 51.0 | 50.6 | 51.4 |
| Min-K%(Shi et al., 2024) | 50.2 | 51.3 | 65.7 | 69.9 | 50.3 | 51.0 | 50.6 | 50.3 |
| Zlib (Carlini et al., 2021) | 51.1 | 52.0 | **67.4** | **71.0** | 49.6 | 50.1 | 49.9 | 50.0 |
| Ref (Carlini et al., 2021) | **51.2** | **55.2** | 63.9 | 67.1 | 49.2 | 52.2 | 51.3 | 53.1 |

| Method | ArXiv 160M | ArXiv 1.4B | DM Math 160M | DM Math 1.4B | HackerNews 160M | HackerNews 1.4B | Average 160M | Average 1.4B |
|---|---|---|---|---|---|---|---|---|
| Infilling Score (Ours) | **51.0** | 51.3 | 53.5 | 50.4 | **50.9** | **52.6** | 53.4 | 54.9 |
| Min-K%++ (Zhang et al., 2024) | 50.1 | 51.1 | 50.5 | 50.9 | 50.7 | 51.3 | 52.4 | 54.1 |
| Min-K%(Shi et al., 2024) | **51.0** | **51.7** | 49.4 | 49.7 | 50.9 | 51.3 | 52.6 | 53.6 |
| Zlib (Carlini et al., 2021) | 50.1 | 50.9 | 48.1 | 48.2 | 49.7 | 50.3 | 52.3 | 53.2 |
| Ref (Carlini et al., 2021) | 49.4 | 51.5 | **51.1** | **51.1** | 49.1 | 52.2 | 52.2 | 54.6 |

Table 3: AUROC results on MIMIR dataset (Duan et al., 2024) for Pythia models for different sizes. Similar to Zhang et al. (2024), we experiment on a subset of MIMIR with maximum 13-gram overlap of 0.8 between samples form "seen" and "unseen" class. Bold shows the best results and underline shows the second best results in each section. As shown, our Infilling Score method overall outperforms existing reference-free and reference-based methods.

| Year Pub. | Book Title | Contamination Rate |
|---|---|---|
| 1817 | Persuasion | 99 |
| 2006 | Oakleaf bearers | 76 |
| 1812 | Grimms' Fairy Tales | 73 |
| 2003 | The Sacred Land | 73 |
| 1986 | Howl's Moving Castle | 69 |
| 2009 | CATCHING FIRE | 68 |
| 1991 | Red Magic | 66 |
| 2009 | Tenth Grade Bleeds | 64 |
| 1998 | Mad Ship | 61 |
| 1996 | Too Good to Leave, Too Bad to Stay | 58 |
| 2009 | Crouching Vampire, Hidden Fang | 56 |
| 1889 | Three Men in a Boat (To Say Nothing of the Dog) | 56 |
| 2003 | Something from the Nightside | 54 |
| 2009 | The Silver Eagle | 53 |
| 1982 | The Man From St. Petersburg | 53 |
| 2000 | Ship of Destiny | 53 |
| 2008 | The Painted Man | 53 |
| 2007 | The Center Cannot Hold | 52 |
| 2007 | Raintree: Sanctuary | 52 |
| 2005 | Sister of the Dead | 52 |
| 2006 | The Corfu Trilogy | 50 |
| 2008 | Ascendancy of the Last | 50 |

Table 4: Books detected in the pretraining data of Llama-3-8B (Dubey et al., 2024). Contamination rate shows the percentage of excerpts sampled from the books which were classified as "seen" using the Infilling Score method.

approach is that it requires testing multiple different LLMs to determine the best performing baseline (Duan et al., 2024; Zhang et al., 2024). Despite the competitive nature of the benchmark, our Infilling Score achieves the best performance compared to both reference-free and reference-based models on average over different domains.

### 4.4.3 DETECTING PRETRAINING DATA OF LLAMA-3

We apply Infilling Score to detect books that were likely used in the pretraining of the Llama3-8B model, recently released by Meta (Dubey et al., 2024). Llama3 is known to be trained using over 15T tokens of data (7x larger training set than Llama-2) according to Dubey et al. (2024). No information about the source and distribution of this data is disclosed by the developers, making it difficult to construct a labeled MIA dataset of books suitable for this model.

We used our books dataset as a validation set to find the best hyperparameters ($k\%$ and # future tokens, $m$, and the classification threshold $\tau$) for identifying samples used in pretraining Llama3. Since Llama3 has been released in 2024, existing temporal benchmarks such as WikiMIA, BookMIA (Shi et al., 2024), and BookTection (Duarte et al., 2024) cannot be used for pretraining data detection on this model. We found that using the next 100 tokens when calculating the Infilling Score shows the highest accuracy on this benchmark. Table 12 shows the performance of our method on this dataset.

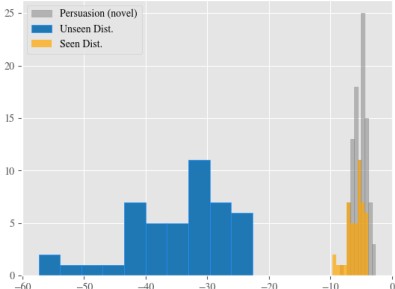

Figure 1: Figure shows an example of the distribution of the Infilling Scores for "seen" and "unseen" excerpts in our validation dataset which consists of text from fiction books. Scores are normalized in each distribution. The unseen data comes from recent novels published after the training of Llama 3. For the classic novel Persuasion, our method detects 99% of the excerpt to be in the training set. As seen in this histogram, the distribution for Persuasion matches other seen novels and is clearly separated from unseen data, as one would expect.

We employ our method on 20,000 excerpts sampled from 200 books. Table 4 presenters the list of books which we found to be in the training dataset of Llama3-8B with $\geq 50\%$ contamination rate. Contamination rate shows the percentage of excerpts detected as 'seen' for each publication. Figure 1 shows that books with high contamination rate have higher sample statistic overlap with the "seen" excerpts in our validation dataset.

### 4.4.4 ABLATION STUDY ON THE NUMBER OF FUTURE TOKENS TO USE

It is important to note that the number of future tokens used to calculate the Infilling Score determines the performance gain of our method. As shown in the Figure 2 increasing the number of future tokens does not necessarily lead to a higher AUC. However, on the WikiMIA benchmark, using about 5 future tokens leads to relatively better AUC across various context lengths on WikiMIA using Llama-7B and Llama-13B. We conduct all experiments with different input sequence lengths (32, 64, 128, and 256) to examine the effect of the number of future tokens across various context lengths. While the ideal number of next tokens to use remains consistent across various model sizes, the optimal number may differ depending on data distribution and model architecture. We investigate various values for $m$ within $\{0, 1, 3, 5, 10, 20, \ldots, N\}$, where $N$ represents the input sequence length. It's important to note that the hyperparameter search does not increase the computational complexity, as incorporating additional future tokens does not require extra calls to the LLM. We provide additional results in Appendix A.

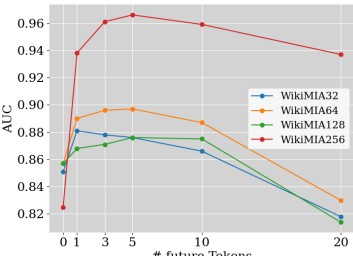
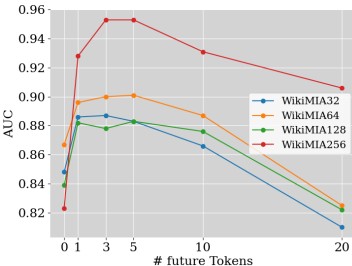

Figure 2: The figures show the AUROC achieved by the Infilling Score as the number of future tokens increases. These results are shown for input sequence lengths of 32, 64, 128, and 256. The left figure presents the results for Llama-7B, while the right figure shows the results for Llama-13B. Our baseline, representing existing methods, uses zero future tokens. The optimal number of future tokens to use is 1 for sequences of 32 tokens. For longer sequences of up to 256 tokens, the optimal number is around 5 for both models.

## 4.5 Algorithm Runtime

Table 5 compares the runtime of our Infilling Score algorithm with straightforward application of Bayes, and Min-K%++ (Shi et al., 2024) using Llama-7B. Although both the naive approach and Infilling score are slower than Min-K%, these methods yield a more accurate estimate of token likelihoods for membership inference. Note that our proposed test statistic, Infilling Score, significantly reduces the computational complexity compared to the naive approach, delivering an accurate membership inference score within a feasible runtime. WikiMIA dataset has 776 sequences of length 32, 542 of length 64, 250 of length 128, and 82 of length 256 tokens. The compute cost increases with the sequence length. The 256-token sequences require approximately 2,460 seconds compared to 776 seconds for 32-token sequences (30 seconds per sequence), highlighting the trade-off between detection accuracy and computational efficiency.

| Seq. length | Min-K%++ | Infilling Score | Naive Approach |
|---|---|---|---|
| 32 | 0.028 sec. | 0.952 sec. | 207 sec. |
| 64 | 0.042 sec. | 3.11 sec. | 334 sec. |
| 128 | 0.064 sec. | 9.47 sec. | 581 sec. |
| 256 | 0.106 sec. | 29.98 sec. | 1141 sec. |

Table 5: Algorithm runtime results comparing Infilling Score, Min-K%++, and the naive approach discussed in Section 3, sequences of 32, 64, 128, and 256 tokens using Llama-7B on a H200 GPU.

To evaluate the impact of the number of future tokens used, $m$, on the runtime, we measure the runtime using 1, 5, and 10 future tokens. As discussed in Section 3.1, the number of LLM calls required by our Infilling Score algorithm is independent of the number of future tokens used. However, increasing the number of future tokens also increases the number of terms in the summations in equation 10. The additional computations have a minimal impact on the runtime as shown in Table 6.

| Seq. length | 32 | | | 64 | | | 128 | | | 256 | | |
|---|---|---|---|---|---|---|---|---|---|---|---|---|
| # future tokens | 1 | 5 | 10 | 1 | 5 | 10 | 1 | 5 | 10 | 1 | 5 | 10 |
| Runtime | 0.952 sec. | 0.953 sec. | 0.956 sec. | 3.11 sec. | 3.12 sec. | 3.12 sec. | 9.47 sec. | 9.48 sec. | 9.49 sec. | 29.98 sec. | 30.01 sec. | 30.04 sec. |

Table 6: Algorithm runtime as the number of future tokens used increases. As the table indicates, increasing the number of future tokens to use has minimal impact on runtime.

## 5 Conclusions

**Limitations** One limitation is that computing the Infilling Score requires grey-box access to the LLM, meaning access to the sample log probabilities estimated by the model. This requirement is common among most of the existing membership inference methods. Another limitation of our approach lies in its runtime complexity. As described in Section 3.1, the order of LLM calls required for computing the infilling likelihood (for a sequence of length $N$) with the naive Bayes method is $N|\mathcal{V}|$, which scales linearly with both sequence length $N$, and vocabulary size $|\mathcal{V}|$. By introducing the Infilling Score, we reduce the number of LLM calls to $2N$. However, prior methods such as Min-K% and Min-K%++ require only a single LLM call (to test a sequence of length $N$), and are faster compared to our proposed algorithm.

To conclude, we proposed a novel method that can detect if text sequences have been present in the training set with significantly better accuracy compared to prior work. Our new test statistic allows us to derive non-causal likelihoods (up to a multiplicative factor) from pre-trained autoregressive models and may have other uses, beyond membership inference. Although our method is slower compared to previous methods, it can be practically run in a few seconds for large foundation models.

Our results present evidence that numerous books and other recent sources of text have been in the training data of modern LLMs. This test can further be used for measuring dataset contamination rates, and also evaluating decontamination methods. An important research direction would be to create larger evaluation datasets for membership inference, and include high n-gram overlap samples for recent sources that remain unseen to llama3 and other recently released frontier models.

## ACKNOWLEDGMENTS

This research has been supported by NSF Grants 2019844, 2112471, AF 1901292, CNS 2148141, Tripods CCF 1934932, Tripods 2217069, NSF AI Institute for Foundations of Machine Learning (IFML) 2019844, the Texas Advanced Computing Center (TACC) and research gifts by Western Digital, Wireless Networking and Communications Group (WNCG) Industrial Affiliates Program, UT Austin Machine Learning Lab (MLL), Cisco and the Stanly P. Finch Centennial Professorship in Engineering.

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

## A  CHOICE OF HYPERPARAMETERS

Infilling Score algorithm has two hyperparameters: $m$ which represents the number of future tokens to use, and $k$ which represents the $k\%$ tokens with minimum probabilities to use. We sweep over $1, 3, 5, 10$ and $20$ future tokens, and $k = 0.1, 0.2, ...0.5$. Tables 7, 8, 9, and 10 show AUROC and TPR at low FPR results on WikiMIA subsets with sequence lengths of 32, 64, 128, and 256. Based on the results, the optimal number of future tokens is 1 for sequences of 32 tokens and 5 for longer sequences. We find that $k = 0.1$ often works best across different model sizes and sequence lengths.

| # future tokens | k (Min-k%) | Llama-7B AUROC | FPR@TPR95 | TPR@FPR05 | Llama-13B AUROC | FPR@TPR95 | TPR@FPR05 | Llama-30B AUROC | FPR@TPR95 | TPR@FPR05 |
|---|---|---|---|---|---|---|---|---|---|---|
| 1 | 0.1 | 89.10 | 33.90 | 30.50 | 89.20 | 32.90 | 24.50 | 87.80 | 37.30 | 27.90 |
| | 0.2 | 88.10 | 36.80 | 27.60 | 88.60 | 35.50 | 26.40 | 87.30 | 37.80 | 27.40 |
| | 0.3 | 88.00 | 37.80 | 27.90 | 88.60 | 36.20 | 26.90 | 87.30 | 39.10 | 28.90 |
| | 0.4 | 88.10 | 36.00 | 25.60 | 88.60 | 35.20 | 27.60 | 87.30 | 37.80 | 28.20 |
| | 0.5 | 88.00 | 37.30 | 25.30 | 88.60 | 36.80 | 25.60 | 87.30 | 38.80 | 27.10 |
| 3 | 0.1 | 88.20 | 34.70 | 29.50 | 89.00 | 32.10 | 29.50 | 86.80 | 36.00 | 27.90 |
| | 0.2 | 87.80 | 35.70 | 31.00 | 88.70 | 38.30 | 28.70 | 86.60 | 37.50 | 27.40 |
| | 0.3 | 87.80 | 37.30 | 31.80 | 88.60 | 38.80 | 28.40 | 86.60 | 37.50 | 28.90 |
| | 0.4 | 87.80 | 36.00 | 32.00 | 88.70 | 38.30 | 30.50 | 86.60 | 37.50 | 27.10 |
| | 0.5 | 87.70 | 37.30 | 28.20 | 88.70 | 38.00 | 30.00 | 86.60 | 36.80 | 28.70 |
| 5 | 0.1 | 88.00 | 35.50 | 34.10 | 88.40 | 32.90 | 27.60 | 87.30 | 35.20 | 31.80 |
| | 0.2 | 87.60 | 37.50 | 32.80 | 88.30 | 33.70 | 29.50 | 87.20 | 35.20 | 33.10 |
| | 0.3 | 87.60 | 37.80 | 32.80 | 88.20 | 36.20 | 27.60 | 87.30 | 35.50 | 31.50 |
| | 0.4 | 87.70 | 37.50 | 33.90 | 88.30 | 35.20 | 30.50 | 87.30 | 35.70 | 32.00 |
| | 0.5 | 87.50 | 37.80 | 30.50 | 88.30 | 34.40 | 28.90 | 87.30 | 35.50 | 31.00 |
| 10 | 0.1 | 86.80 | 39.60 | 28.40 | 86.80 | 37.00 | 26.90 | 85.30 | 39.80 | 28.40 |
| | 0.2 | 86.60 | 41.10 | 26.60 | 86.60 | 37.50 | 26.90 | 85.20 | 40.60 | 27.90 |
| | 0.3 | 86.60 | 41.40 | 27.10 | 86.60 | 36.20 | 26.40 | 85.20 | 40.90 | 27.90 |
| | 0.4 | 86.60 | 41.60 | 26.90 | 86.70 | 35.50 | 26.10 | 85.20 | 41.40 | 27.90 |
| | 0.5 | 86.40 | 41.60 | 25.10 | 86.60 | 36.00 | 26.40 | 85.30 | 40.90 | 27.90 |
| 20 | 0.1 | 82.00 | 47.30 | 22.70 | 81.20 | 44.20 | 19.60 | 81.30 | 47.80 | 22.20 |
| | 0.2 | 81.80 | 47.80 | 23.50 | 81.00 | 45.50 | 20.20 | 81.20 | 49.10 | 22.20 |
| | 0.3 | 81.80 | 47.80 | 22.50 | 80.90 | 44.50 | 19.90 | 81.10 | 49.40 | 22.20 |
| | 0.4 | 81.80 | 48.10 | 23.00 | 81.00 | 45.80 | 20.20 | 81.20 | 48.60 | 22.20 |
| | 0.5 | 81.40 | 46.80 | 21.70 | 80.80 | 45.00 | 20.20 | 81.20 | 48.30 | 22.20 |

Table 7: Complete Infilling Score results testing Llama-7B, Llama-13B, and Llama-30B models on the Original subset of the WikiMIA 32-token sequences (Shi et al., 2024). For this subset, using one future token results in the best performance.

| # future tokens | k (Min-k%) | Llama-7B AUROC | FPR@TPR95 | TPR@FPR05 | Llama-13B AUROC | FPR@TPR95 | TPR@FPR05 | Llama-30B AUROC | FPR@TPR95 | TPR@FPR05 |
|---|---|---|---|---|---|---|---|---|---|---|
| 1 | 0.1 | 89.60 | 33.70 | 44.00 | 89.90 | 35.30 | 46.10 | 87.70 | 36.40 | 39.10 |
| | 0.2 | 89.00 | 36.40 | 45.40 | 89.60 | 38.40 | 47.90 | 87.50 | 35.70 | 38.40 |
| | 0.3 | 89.00 | 35.70 | 45.80 | 89.60 | 39.50 | 47.20 | 87.50 | 35.70 | 39.40 |
| | 0.4 | 89.00 | 36.80 | 46.80 | 89.60 | 37.60 | 47.20 | 87.60 | 35.70 | 38.70 |
| | 0.5 | 89.00 | 36.40 | 45.80 | 89.60 | 39.50 | 47.20 | 87.60 | 35.70 | 39.10 |
| 3 | 0.1 | 89.70 | 37.60 | 41.20 | 90.00 | 36.80 | 52.10 | 88.00 | 37.60 | 36.30 |
| | 0.2 | 89.60 | 38.00 | 38.70 | 90.00 | 37.60 | 53.50 | 88.00 | 35.70 | 37.30 |
| | 0.3 | 89.60 | 37.60 | 37.30 | 90.00 | 38.80 | 52.50 | 88.00 | 35.70 | 37.30 |
| | 0.4 | 89.70 | 37.60 | 39.80 | 90.00 | 38.40 | 53.50 | 88.00 | 35.70 | 37.70 |
| | 0.5 | 89.70 | 37.60 | 39.40 | 90.00 | 38.40 | 53.20 | 88.00 | 35.70 | 37.70 |
| 5 | 0.1 | 89.70 | 38.00 | 45.80 | 90.10 | 37.20 | 48.60 | 88.30 | 35.30 | 43.00 |
| | 0.2 | 89.70 | 37.20 | 45.40 | 90.10 | 39.50 | 49.30 | 88.30 | 34.90 | 43.70 |
| | 0.3 | 89.70 | 38.40 | 45.80 | 90.10 | 39.10 | 50.00 | 88.30 | 34.10 | 44.00 |
| | 0.4 | 89.70 | 37.20 | 46.10 | 90.10 | 39.10 | 49.60 | 88.40 | 35.30 | 44.00 |
| | 0.5 | 89.70 | 38.00 | 46.50 | 90.00 | 38.80 | 46.10 | 88.30 | 35.30 | 43.70 |
| 10 | 0.1 | 88.70 | 40.30 | 52.80 | 88.70 | 40.70 | 47.50 | 87.30 | 38.80 | 37.70 |
| | 0.2 | 88.70 | 41.10 | 52.50 | 88.70 | 41.50 | 47.20 | 87.30 | 38.00 | 38.70 |
| | 0.3 | 88.70 | 41.10 | 52.10 | 88.70 | 41.50 | 47.50 | 87.30 | 38.00 | 38.70 |
| | 0.4 | 88.70 | 40.70 | 53.20 | 88.80 | 41.90 | 47.50 | 87.30 | 38.40 | 39.10 |
| | 0.5 | 88.60 | 41.10 | 53.50 | 88.50 | 42.20 | 46.80 | 87.30 | 37.60 | 38.00 |
| 20 | 0.1 | 83.00 | 51.90 | 33.10 | 82.50 | 51.20 | 26.40 | 82.50 | 49.60 | 36.30 |
| | 0.2 | 83.00 | 52.70 | 32.40 | 82.50 | 50.80 | 27.10 | 82.50 | 49.60 | 35.60 |
| | 0.3 | 82.90 | 52.70 | 32.70 | 82.50 | 50.80 | 27.10 | 82.50 | 49.20 | 35.90 |
| | 0.4 | 83.00 | 52.30 | 32.40 | 82.50 | 50.80 | 27.50 | 82.50 | 49.20 | 35.60 |
| | 0.5 | 82.90 | 52.70 | 33.10 | 82.30 | 50.80 | 27.10 | 82.40 | 48.80 | 35.60 |

Table 8: Complete Infilling Score results testing Llama-7B, Llama-13B, and Llama-30B models on the Original subset of the WikiMIA 64-token sequences (Shi et al., 2024). For this subset, using five future tokens results in the best performance.

| # future tokens | k (Min-k%) | Llama-7B | | | Llama-13B | | | Llama-30B | | |
|---|---|---|---|---|---|---|---|---|---|---|
| | | AUROC | FPR@TPR95 | TPR@FPR05 | AUROC | FPR@TPR95 | TPR@FPR05 | AUROC | FPR@TPR95 | TPR@FPR05 |
| 1 | 0.1 | 87.10 | 36.90 | 36.00 | 88.40 | 35.10 | 40.30 | 84.90 | 33.30 | 24.50 |
| | 0.2 | 86.80 | 36.90 | 35.30 | 88.20 | 34.20 | 39.60 | 84.70 | 36.90 | 24.50 |
| | 0.3 | 86.80 | 36.00 | 35.30 | 88.20 | 34.20 | 40.30 | 84.70 | 36.90 | 24.50 |
| | 0.4 | 86.80 | 36.90 | 36.00 | 88.20 | 34.20 | 39.60 | 84.70 | 37.80 | 24.50 |
| | 0.5 | 86.80 | 36.90 | 36.70 | 88.20 | 34.20 | 41.00 | 84.70 | 36.90 | 24.50 |
| 3 | 0.1 | 87.20 | 37.80 | 22.30 | 87.90 | 36.00 | 36.70 | 85.30 | 39.60 | 13.70 |
| | 0.2 | 87.10 | 37.80 | 23.70 | 87.80 | 34.20 | 37.40 | 85.20 | 39.60 | 14.40 |
| | 0.3 | 87.10 | 37.80 | 23.00 | 87.80 | 34.20 | 37.40 | 85.20 | 39.60 | 13.70 |
| | 0.4 | 87.10 | 37.80 | 23.70 | 87.80 | 35.10 | 37.40 | 85.20 | 39.60 | 13.70 |
| | 0.5 | 87.10 | 37.80 | 22.30 | 87.80 | 34.20 | 37.40 | 85.20 | 39.60 | 13.70 |
| 5 | 0.1 | 87.70 | 38.70 | 37.40 | 88.40 | 37.80 | 34.50 | 86.70 | 37.80 | 19.40 |
| | 0.2 | 87.60 | 38.70 | 38.10 | 88.30 | 37.80 | 33.80 | 86.70 | 37.80 | 20.10 |
| | 0.3 | 87.60 | 38.70 | 36.70 | 88.30 | 37.80 | 34.50 | 86.60 | 37.80 | 19.40 |
| | 0.4 | 87.60 | 38.70 | 37.40 | 88.30 | 37.80 | 34.50 | 86.60 | 37.80 | 19.40 |
| | 0.5 | 87.60 | 38.70 | 36.70 | 88.30 | 36.90 | 35.30 | 86.60 | 37.80 | 20.10 |
| 10 | 0.1 | 87.60 | 45.90 | 23.70 | 87.60 | 39.60 | 33.80 | 86.00 | 38.70 | 18.00 |
| | 0.2 | 87.50 | 44.10 | 24.50 | 87.60 | 40.50 | 34.50 | 86.00 | 38.70 | 18.00 |
| | 0.3 | 87.60 | 43.20 | 23.70 | 87.50 | 40.50 | 35.30 | 85.90 | 38.70 | 18.00 |
| | 0.4 | 87.50 | 44.10 | 23.70 | 87.50 | 39.60 | 35.30 | 85.90 | 37.80 | 18.00 |
| | 0.5 | 87.50 | 43.20 | 23.70 | 87.50 | 39.60 | 33.80 | 85.90 | 38.70 | 18.00 |
| 20 | 0.1 | 81.50 | 54.10 | 21.60 | 82.30 | 53.20 | 18.70 | 83.20 | 52.30 | 20.90 |
| | 0.2 | 81.40 | 55.00 | 23.00 | 82.20 | 52.30 | 18.70 | 83.10 | 51.40 | 20.90 |
| | 0.3 | 81.40 | 55.00 | 21.60 | 82.20 | 52.30 | 18.70 | 83.10 | 51.40 | 21.60 |
| | 0.4 | 81.40 | 55.00 | 21.60 | 82.20 | 52.30 | 18.70 | 83.20 | 51.40 | 20.90 |
| | 0.5 | 81.40 | 55.00 | 21.60 | 82.10 | 52.30 | 18.70 | 83.10 | 51.40 | 20.90 |

Table 9: Complete Infilling Score results testing Llama-7B, Llama-13B, and Llama-30B models on the Original subset of the WikiMIA 128-token sequences (Shi et al., 2024). Again, using five future tokens results in the best performance.

| # future tokens | k (Min-k%) | Llama-7B | | | Llama-13B | | | Llama-30B | | |
|---|---|---|---|---|---|---|---|---|---|---|
| | | AUROC | FPR@TPR95 | TPR@FPR05 | AUROC | FPR@TPR95 | TPR@FPR05 | AUROC | FPR@TPR95 | TPR@FPR05 |
| 1 | 0.1 | 93.80 | 51.60 | 80.40 | 92.90 | 25.80 | 66.70 | 85.60 | 32.30 | 21.60 |
| | 0.2 | 93.80 | 51.60 | 80.40 | 92.80 | 29.00 | 68.60 | 85.60 | 35.50 | 21.60 |
| | 0.3 | 93.70 | 51.60 | 80.40 | 92.70 | 29.00 | 66.70 | 85.60 | 32.30 | 19.60 |
| | 0.4 | 93.80 | 51.60 | 80.40 | 92.70 | 29.00 | 66.70 | 85.60 | 35.50 | 21.60 |
| | 0.5 | 93.90 | 51.60 | 80.40 | 92.70 | 29.00 | 66.70 | 85.70 | 32.30 | 19.60 |
| 3 | 0.1 | 96.30 | 29.00 | 78.40 | 95.30 | 19.40 | 72.50 | 90.60 | 41.90 | 72.50 |
| | 0.2 | 96.10 | 32.30 | 74.50 | 95.30 | 19.40 | 72.50 | 90.60 | 41.90 | 72.50 |
| | 0.3 | 96.00 | 35.50 | 74.50 | 95.30 | 19.40 | 72.50 | 90.60 | 41.90 | 72.50 |
| | 0.4 | 96.00 | 35.50 | 74.50 | 95.30 | 19.40 | 72.50 | 90.60 | 41.90 | 72.50 |
| | 0.5 | 96.00 | 35.50 | 72.50 | 95.30 | 19.40 | 72.50 | 90.60 | 41.90 | 72.50 |
| 5 | 0.1 | 96.80 | 22.60 | 74.50 | 95.30 | 22.60 | 80.40 | 89.80 | 35.50 | 47.10 |
| | 0.2 | 96.60 | 22.60 | 74.50 | 95.30 | 22.60 | 80.40 | 89.80 | 35.50 | 47.10 |
| | 0.3 | 96.50 | 25.80 | 74.50 | 95.20 | 22.60 | 80.40 | 89.80 | 35.50 | 47.10 |
| | 0.4 | 96.60 | 22.60 | 74.50 | 95.20 | 22.60 | 80.40 | 89.80 | 35.50 | 47.10 |
| | 0.5 | 96.60 | 25.80 | 74.50 | 95.20 | 22.60 | 80.40 | 89.80 | 35.50 | 47.10 |
| 10 | 0.1 | 95.70 | 29.00 | 78.40 | 93.00 | 29.00 | 54.90 | 87.40 | 45.20 | 49.00 |
| | 0.2 | 95.90 | 25.80 | 78.40 | 93.10 | 29.00 | 54.90 | 87.50 | 45.20 | 49.00 |
| | 0.3 | 95.80 | 29.00 | 78.40 | 93.20 | 29.00 | 54.90 | 87.60 | 45.20 | 49.00 |
| | 0.4 | 95.80 | 29.00 | 78.40 | 93.10 | 29.00 | 54.90 | 87.50 | 45.20 | 49.00 |
| | 0.5 | 95.80 | 29.00 | 78.40 | 93.20 | 29.00 | 54.90 | 87.50 | 45.20 | 49.00 |
| 20 | 0.1 | 93.70 | 22.60 | 66.70 | 90.70 | 35.50 | 51.00 | 85.60 | 48.40 | 35.30 |
| | 0.2 | 93.70 | 22.60 | 66.70 | 90.60 | 35.50 | 51.00 | 85.60 | 48.40 | 33.30 |
| | 0.3 | 93.40 | 22.60 | 68.60 | 90.60 | 35.50 | 51.00 | 85.80 | 48.40 | 35.30 |
| | 0.4 | 93.70 | 22.60 | 68.60 | 90.60 | 35.50 | 51.00 | 85.70 | 48.40 | 35.30 |
| | 0.5 | 93.50 | 22.60 | 66.70 | 90.50 | 35.50 | 51.00 | 85.70 | 48.40 | 35.30 |

Table 10: Complete Infilling Score results testing Llama-7B, Llama-13B, and Llama-30B models on the Original subset of the WikiMIA 256-token sequences (Shi et al., 2024). Similar to the WikiMIA 64-token and 128-token sequence subsets, using 5 future tokens results in the best performance.

| Sequence Length | Model | Infilling Score | | Min-K%++ | | Comparison | |
|---|---|---|---|---|---|---|---|
| | | AUROC (%) | Std Err | AUROC (%) | Std Err | Difference (%) | p-value |
| 32 tokens | llama-7b | 89.185 | 1.173 | 85.182 | 1.328 | 4.003 ± 1.130 | 0.000*** |
| | llama-13b | 88.850 | 1.232 | 84.852 | 1.333 | 3.998 ± 1.222 | 0.004** |
| | llama-30b | 87.628 | 1.236 | 84.390 | 1.329 | 3.239 ± 1.157 | 0.006** |
| 64 tokens | llama-7b | 89.788 | 1.341 | 85.922 | 1.659 | 3.866 ± 1.492 | 0.012* |
| | llama-13b | 90.029 | 1.265 | 85.692 | 1.642 | 4.338 ± 1.539 | 0.010* |
| | llama-30b | 88.206 | 1.447 | 84.828 | 1.705 | 3.378 ± 1.601 | 0.040* |
| 128 tokens | llama-7b | 87.364 | 2.272 | 84.896 | 2.395 | 2.468 ± 2.654 | 0.348 |
| | llama-13b | 88.145 | 2.214 | 83.740 | 2.463 | 4.405 ± 2.649 | 0.080 |
| | llama-30b | 86.207 | 2.797 | 82.398 | 2.602 | 3.809 ± 1.993 | 0.064 |
| 256 tokens | llama-7b | 96.307 | 1.761 | 82.354 | 4.662 | 13.952 ± 4.296 | 0.000*** |
| | llama-13b | 95.124 | 2.271 | 82.326 | 4.740 | 12.797 ± 3.952 | 0.000*** |
| | llama-30b | 90.737 | 3.782 | 77.411 | 5.643 | 13.326 ± 4.459 | 0.002** |

Table 11: Comparing performance of Infilling Score versus Min-K%++ across different sequence lengths and model sizes. Results show bootstrap estimates with 1000 iterations. The mean difference indicates Infilling Score's improvement over Min-K%++. Statistical significance is denoted as: * ($p < 0.05$), ** ($p < 0.01$), *** ($p < 0.001$).

# B    ADDITIONAL RESULTS

## B.1    STATISTICAL ANALYSIS: INFILLING SCORE VS. MIN-K%++

We employ a bootstrap-based statistical comparison to evaluate INFILLING SCORE and MIN-K%++. We use 1,000 bootstrap iterations to estimate the the mean difference between AUROC metrics from these methods, along with the standard errors to construct $95\%$ confidence intervals for the true performance gap. TABLE 11 shows that INFILLING SCORE consistently outperforms MIN-K%++ across different sequence lengths (32, 64, 128, and 256 tokens) and model sizes (7B, 13B, and 30B parameters).

## B.2    DETECTING PRE-TRAINING DATA FROM BOOKS

We compare the AUROC of Infilling Score with existing methods on a labeled validation subset of book excerpts. As discussed in Section 4.1, this validation subset contains book excerpts labeled as "seen" and "unseen". Infilling Score significantly outperforms existing methods in detecting "seen" examples.

| Method | AUC |
|---|---|
| Infilling Score (Ours) | 0.79 |
| Min-K%++ (Zhang et al., 2024) | 0.53 |
| Min-K%(Shi et al., 2024) | 0.71 |
| Zlib (Carlini et al., 2021) | 0.68 |

Table 12: Comparing AUROC of Infilling Score, Min-K%++, Min-K%, and Zlib methods on the validation dataset, detecting book excerpts in Llama3-8B pretraining data.

# C    COMPUTE RESOURCES

We ran our experiments on A100 (40 GB) and H200 (120 GB) GPUs. Testing Infilling Score on the WikiMIA benchmark on an A100 node takes approximately between 20 minutes (for a 3B parameter model) and 35 minutes (for a 30B parameter model). For Llama models, we used float16 data type. On the MIMIR benchmark, where there are 1000 long samples per class, the test approximately takes 10 hours on each subset on an A100 node.

# D  INFILLING SCORE ALGORITHM

---

**Algorithm 1:** Infilling Score

---

**Input:** Sequence: $x : x_1, x_2 \ldots x_N$, Threshold $\tau$

**1** **for** $i = 1$ **to** $N$ **do**

**2** $\quad$ Compute $\log p(x_i | x_1 \ldots x_{i-1})$

**3** $\quad \mu_{x_{<i}} \leftarrow \mathbb{E}_{z \sim p(\cdot | x_1 \ldots x_{i-1})}[\log p(z | x_1 \ldots x_{i-1})]$

**4** $\quad \sigma_{x_{<i}} \leftarrow \sqrt{\mathbb{E}_{z \sim p(\cdot | x_1 \ldots x_{i-1})}[(\log p(z | x_1 \ldots x_{i-1}) - \mu_{x_{<i}})^2]}$

**5** $\quad$ Find $x_i^* \leftarrow \arg\max_{x_i' \in \mathcal{V}} p(x_i' | x_1 \ldots x_{i-1})$

**6** $\quad$ Compute $\log p(x_i^* | x_1 \ldots x_{i-1})$

**7** $\quad r \leftarrow (\log p(x_i | x_1 \ldots x_{i-1}) - \mu_{x_{<i}}) / \sigma_{x_{<i}} - (\log p(x_i^* | x_1 \ldots x_{i-1}) - \mu_{x_{<i}}) / \sigma_{x_{<i}}$

**8** $\quad$ **for** $j = i + 1$ **to** $i + m$ **do**

**9** $\quad\quad$ Compute $\log p(x_j | x_1 \ldots x_{j-1})$

**10** $\quad\quad$ Compute $\log p(x_j | x_1 \ldots x_{i*} \ldots x_{j-1})$

**11** $\quad\quad r \leftarrow r + (\log p(x_j | x_1 \ldots x_{j-1}) - \mu_{x<i}) / \sigma_{x<i} - (\log p(x_j | x_1 \ldots x_{i*} \ldots x_{j-1}) - \mu_{x<i}) / \sigma_{x<i}$

**12** $\quad$ **end**

**13** $\quad$ InfillingScore$_{\text{token}}(x_i) \leftarrow r$

**14** **end**

**15** Min-K%$(x) \leftarrow$ k% of tokens from $x$ with the lowest InfillingScore$_{\text{token}}(x_i)$

**16** InfillingScore$(\boldsymbol{x}) = \sum_{x_i \in \text{min-}k\%}$ InfillingScore$_{\text{token}}(x_i)$

**17** **return** InfillingScore$(\boldsymbol{x}) < \tau$

---

