# OpenReview forum: "Infilling Score: A Pretraining Data Detection Algorithm for Large Language Models"
_ICLR.cc/2025/Conference — ICLR 2025 Poster_

### Official Review · Reviewer_T2Q9 · 2024-11-02

**Soundness:** 4
**Presentation:** 3
**Contribution:** 3
**Rating:** 6
**Confidence:** 3

**Summary:**

This paper studies algorithms to detect whether a given sentence is the pretraining data used for training an LLM. The paper proposes “Infilling Score”, a test-statistic score based on non-causal token likelihoods. The proposed score works by computing the infilling probability of a token based on both its past and future tokens, whereas similar existing methods, such as MIN-K% and MIN-K%++, rely on a statistic based on past tokens only.

The motivation for Infilling Score is that incorporating future tokens should yield a more indicative measure. While the most intuitive method to incorporate future tokens is based on the Bayes rule, the paper argues that it would involve marginalising over all possible tokens in the vocabulary at each time step (i.e., each token in the test sequence). Therefore, this paper proposes an approximation (in Equation 6) to circumvent the expensive marginalisation, requiring 2 * the number of test tokens calls instead of the number of vocab size * the number of test tokens calls.

Experiments are conducted on standard datasets, including WikiMIA and MIMIR, and the proposed method is compared against other reference-free methods (e.g., Min-K% and Min-K%++) as well as other reference-based methods. Experimental results show that Infilling Score outperforms existing methods in the majority of tasks. In addition, the paper complies a more recent data from book excerpts and applies the detection algorithm on Llama3.

**Strengths:**

- The idea is simple, yet novel and effective. The method extends the existing reference-free pretraining data detection algorithm (Min-K++%) by incorporating the information from future tokens, and the proposed method yields improved performance.
- While it is not as efficient as Min-K%, the proposed algorithm is more efficient than the implementation of the naive Bayes rules. This is noted and examined in the paper.
- Comprehensive experiments on both standard datasets as well as a case study on new datasets (to avoid likely contamination).

Overall, this paper improves the understanding in the field of contamination/training data detection.

**Weaknesses:**

- It is not clear in Section 4 (experiments) how Infilling Score (Equation 6) performs compared to the naive Bayes rule (Equation 5). The paper provides run-time in Section 4.5, but I cannot find the performance comparison.
- [nitpick] similar to Min-K% & Min-K%++, this method is limited to grey-box scenarios.

**Questions:**

In the of data contamination detection, I wonder how this method works if the example to be tested (x) is partially contaminated. Would it be more or less sensitive to partial changes compared to other methods?

---

> ### Author Response · Authors · 2024-11-21
>
> Thank you so much for your review, we appreciate the points you have brought up.
>
> ### W1.
> About the weakness point 1, where you have asked about performance comparison of the InfillingScore methods vs. the exact score computation:
> Thanks for bringing up this point. It definitely would have been ideal to compare the accuracies of exact score computation, against our proposed method. However, it is not feasible to run, as it is extremely compute and runtime intensive to run the naive approach on the entire dataset. Based on the runtime provided in Section 4.5, it takes approximately a couple of days to run the tests for a single subset of WikiMIA. Hence, this was infeasible for us.
>
> ### W2.
> About the weakness point 2, where you have mentioned grey-box access limitation:
> This is correct. However, it is important to note that a majority of the foundational LLMs in industry such as Llama, Mistral, Qwen, and Gemma, are open-weights models where can be tested with these methods.
>
> ### Q1.
> About the question you have brought up:
> This is very interesting comparison to do. Can you elaborate a bit more on the “partial contamination” setting? Do you mean a setting where only a portion of a sentence tokens were seen by the model during training?
> The benefit of our method compared to MinK% and MinK%++ methods is that InfillingScore incorporates future tokens as well as part tokens for membership detection. So, my prediction is that this advantage still holds under noise.
> If your suggestion is to test and compare a setting where a only a subset or a random set of tokens have been seen by the model during training, we should be able to test that by randomly changing a subset of tokens from WikiMIA samples and compare how InfillingScore performs on this noisy dataset compared to MinK%++.

---

> > ### Comment · Reviewer_T2Q9 · 2024-11-25
> >
> > thank you for the response:
> >
> > Regarding Weakness 1, I understand your point. Sometimes, it could be useful to report the performance on a smaller subset ofthe  dataset (subject to the compute resource).
> >
> > Regarding the question, yes you are correct. That was my interpretation. Thank you for your explanation anyway.

---

### Official Review · Reviewer_7osK · 2024-11-04

**Soundness:** 4
**Presentation:** 4
**Contribution:** 4
**Rating:** 8
**Confidence:** 4

**Summary:**

This paper proposed a pretraining data detection approach that utilizes the non-causal token likelihoods which depend on both preceding and succeeding token probabilities.

**Strengths:**

1. The paper has enough novelty.
2. It includes all the previously related works and lists the differences.
3. It has detailed experiments and results analysis.
4. The paper writing is clear, and the visuals are good.

**Weaknesses:**

The proposed approach has a slower speed compared to the previous best approach. However, I believe it has little effect on the impact because contamination detection may not need to be so fast.

**Questions:**

N/A

---

> ### Author Response · Authors · 2024-11-21
>
> Thank you for your review. We are glad and encouraged that you found InfillingScore a novel method.
>
> On the weakness point you have mentioned:
> Yes, thanks for highlighting this very important point. The first step in evaluating a model with InfillingScore on a corpus is to determine the optimal classification threshold. This can be achieved using 100 positive (seen) and 100 negative (unseen) sample sentences, requiring approximately 1.5 hours on H200 GPU nodes with Llama3-8B. Importantly, this process of determining the threshold is a one-time process per model, ensuring runtime is feasible and InfillingScore is practical for testing purposes. The second step, which is testing the samples, takes tens of seconds for a typical sentence of 100-200 tokens, which is feasible for membership inference purposes.

---

> > ### Comment · Reviewer_7osK · 2024-11-26
> >
> > Thank you for the replies! I will keep the score.

---

### Official Review · Reviewer_ZiHR · 2024-11-04

**Soundness:** 3
**Presentation:** 3
**Contribution:** 3
**Rating:** 8
**Confidence:** 3

**Summary:**

In pre-training data detection, this paper proposed Infilling Score, a new test-statistic based on non-causal token likelihood. This paper also proposed a ratio test-statistic whose computation is invariant to vocabulary size. Experiments showed the method achieves an accuracy gain over state-of-the-art methods, including Min-K%, and Min-K%++ on the WikiMIA benchmark. Finally, the paper constructed a benchmark dataset that consists of recent data sources published after the release of Llama-3; this benchmark can provide a statistical baseline to indicate potential corpora used for Llama-3 training.

**Strengths:**

It is natural but good idea to introduce a new test-statistic that is based on non-causal token likelihood. Empirically, the method achieved an accuracy gain over state-of-the-art methods.

**Weaknesses:**

1. While the method empirically achieved an accuracy gain over state-of-the-art methods in Tables 1 and 3, I wonder the differences are really significant. If so, it is better to mention in what significance test the authors found they are significant.

2. While the computation can be invariant to the vocabulary size in the proposed method, it seems that there exists a tradeoff in terms of the sequence length. The performance is lower for shorter instances than longer ones. However, for longer instances, the computation is higher. In Sec. 4.5, since the authors did not show the size of the datasets with different lengths, it is difficult to judge whether the differences in the speed in Table 5 are negligible or not. It is better to clearly mention the number of instances for each dataset.

It is more convincing when, if possible, the authors can try to estimate the runtime for actual settings where we try pre-training data detection for a specific dataset of a certain size, rather than for the standard benchmark dataset.

**Questions:**

1. I wonder \tau in Sec. 3.2 is also a hyperparameter to be fixed. The value and how the authors fixed it were not mentioned in the experiments  Please explain them.

---

> ### Author Response · Authors · 2024-11-23
>
> Thanks you for your review, we appreciate the great suggestions and feedback.
>
> ### W1.
> Thank you for your great suggestion: we run a bootstrap hypothesis testing and calculate the two-sided p-value, comparing Infilling Score AUROC results with Mink%++ results.
> We have updated the paper with these results in Appendix B.1, Table 11.
> Here's the table converted to markdown format:
>
> | Sequence Length | Model | Infilling Score |  | MinK++ |  | Comparison |  |
> | --- | --- | --- | --- | --- | --- | --- | --- |
> |  |  | AUROC (%) | Std Err | AUROC (%) | Std Err | Difference (%) | p-value |
> | 32 tokens | llama-7b | 89.185 | 1.173 | 85.182 | 1.328 | 4.003 ± 1.130 | 0.000*** |
> |  | llama-13b | 88.850 | 1.232 | 84.852 | 1.333 | 3.998 ± 1.222 | 0.004** |
> |  | llama-30b | 87.628 | 1.236 | 84.390 | 1.329 | 3.239 ± 1.157 | 0.006** |
> | 64 tokens | llama-7b | 89.788 | 1.341 | 85.922 | 1.659 | 3.866 ± 1.492 | 0.012* |
> |  | llama-13b | 90.029 | 1.265 | 85.692 | 1.642 | 4.338 ± 1.539 | 0.010* |
> |  | llama-30b | 88.206 | 1.447 | 84.828 | 1.705 | 3.378 ± 1.601 | 0.040* |
> | 128 tokens | llama-7b | 87.364 | 2.272 | 84.896 | 2.395 | 2.468 ± 2.654 | 0.348 |
> |  | llama-13b | 88.145 | 2.214 | 83.740 | 2.463 | 4.405 ± 2.649 | 0.080 |
> |  | llama-30b | 86.207 | 2.797 | 82.398 | 2.602 | 3.809 ± 1.993 | 0.064 |
> | 256 tokens | llama-7b | 96.307 | 1.761 | 82.354 | 4.662 | 13.952 ± 4.296 | 0.000*** |
> |  | llama-13b | 95.124 | 2.271 | 82.326 | 4.740 | 12.797 ± 3.952 | 0.000*** |
> |  | llama-30b | 90.737 | 3.782 | 77.411 | 5.643 | 13.326 ± 4.459 | 0.002** |
>
> Results show bootstrap estimates with 1000 iterations. The mean difference indicates Infilling Score's improvement over MinK++. Statistical significance is denoted as: * (p < 0.05), ** (p < 0.01), *** (p < 0.001).
>
> ### W2.
> Thanks for bringing up this point. The dataset has 776 sequences of length 32, 542 sequences of length 64, 250 sequences of length 128, and 82 sequences of length 256. Hence, there is a trade-off between accuracy and runtime, as testing the 256-token sequences takes about 82 x 30 = 2460 sec (vs. 776 sec for 32-token sequences).
>
> Note that it is still quite feasible to test the dataset on a single GPU node.
> We have added this explanation in the paper.
>
> Also, the process of testing an actual dataset (with no labels) has two steps. The first step is detecting the optimum classification threshold, where one can use 100 positive (seen) and 100 negative (unseen) sample sentences to choose the optimum threshold.
> A book or a corpus of 100,000 words, or ~75,000 tokens, when split into 256-token chunks, takes (75000 / 256) * 30 seconds, or ~ 2.5 hrs to test fully using InfillingScore. Note that in many cases, we can randomly sample a number of paragraphs from a corpus to test, and calculate the contamination rate (example in: Shi. et. al., Table 2. https://arxiv.org/pdf/2310.16789)
>
> ### Q1.
> Thanks for mentioning this, indeed determining this parameter can cause confusion. The parameter τ is the classification threshold which changes based on the data distribution. To find the optimum classification threshold, the process is to construct labeled positive (seen) and negative (unseen) subsets, and use the sample scores to find the best threshold based on accuracy or AUROC. We added the explanation for this in lines 131-134 in the paper.

---

> > ### Comment · Reviewer_ZiHR · 2024-11-26
> >
> > Thank you for replying to my comments. These addressed my concerns. As for the last part in W2, I suggest you to add the explanation in your final version of the paper.

---

> > > ### Author Response · Authors · 2024-11-28
> > >
> > > Thanks for your valuable feedback, we have updated the paper.

---

### Official Review · Reviewer_kRF2 · 2024-11-08

**Soundness:** 2
**Presentation:** 2
**Contribution:** 2
**Rating:** 3
**Confidence:** 4

**Summary:**

The paper introduces the Infilling Score, a method developed based on Min-k% and Min-k%++ to detect whether a given text sequence was part of a language model’s pretraining data, which is the traditional MIA setting. This method builds on existing approaches for membership inference attacks by using non-causal token likelihoods to improve detection accuracy. The authors propose a ratio test-statistic for efficient computation, and demonstrate its effectiveness on various benchmarks such as WikiMIA and MIMIR, where the proposed method performs slightly better than the compared baselines, especially on longer text sequences.

**Strengths:**

1. The motivation of combining the exisitng method and likelihood probability is straightforward and easy to understand.
2. This paper presents extensive experiments to demonstrate the effectiveness of the proposed method. And the case study with the newly-released model Llama 3 could be useful for real practices.

**Weaknesses:**

1. The contribution in this paper is quite marginal. The propose framework is more like an extension of the previous methods like Min-k% and Min-k%++, which utilizes token probabilities to make predictions. The token probabilities approach have been shown to be less effective for white-box settings in MIA studies. While I understand the grey-box access might raise more challenges, the modifications of previous methods in this paper can hardly contribute to new insights into this direction.

2. It seems that the complexity of the proposed framework is proportional to the number of tokens considered in the method. As shown in Sec. 4.5, the runtime of the proposed method is indeed much higher than the previous Min-k%++ methods. Would this become a concern when using such method in real practices?

3. Despite the increased complexity, the improvements over the baseline models in terms of performance in these benchmarkes are also not obvious. In many cases, Infiling Score can only achieve similar performances with Min-k% and Min-k%++.

**Questions:**

See the weaknesses above.

---

> ### Author Response · Authors · 2024-11-21
>
> ### W1. Marginal Contribution
> Thanks for your review. We want to clarify what are contributions are:
> - The first contribution is the introduction of the infilling score metric and showing that the involved Bayesian probabilities significantly improve membership performance compared to methods like MinK% and MinK%++ which only rely on past tokens. This new statistic is an improvement over the previous state of the art methods, so it is non-trivial. The problem is that the computation of the Exact Infilling score is very expensive: It requires a number of LLM calls proportional to the vocabulary size. Roughly, vocabulary sizes are typically 30-50k tokens and an exact calculation of the infilling score would require 100-200k LLM calls.
> - The second key contribution is algorithmic: how to approximate the infilling score with only two LLM calls, as opposed to thousands. The technical innovation of our method is how to make our approximation algorithm not depend on the vocabulary size |V|. This is a key that we don’t want our reviewers to miss.
>
> Also, InfillingScore performs significantly better than Min-K% and Min-K++ on LLMs with larger pretraining datasets, such as LLama. This is particularly valuable because as the size of the training corpus grows, it is more likely to be held internally as proprietary data rather than being publicly shared.
>
> ### W2. Complexity Concerns
> Although InfillingScore has a longer runtime compared to other inference-based methods like MinK% and MinK%++, it is important to note that this method is still faster and cheaper than training data detection methods that require further model training (example: Zhang et. al., https://www.arxiv.org/pdf/2410.10880)
> The process of detecting the best classification threshold needs testing with two subsets of seen and unseen examples as our groundtruth (labeled) dataset. Using 100 sample sentences (of 128 to 256 tokens) per class, this process takes less than 1.5 hour for the Llama-7B model on one H200 GPU. Noe that this is a one-time process. Once the optimum threshold is detected, testing a paragraph of text takes tens of seconds in this setting.
>
> ### W3. Performance Improvements
> Response:
> InfillingScore significantly outperforms MinK% and MinK%++  on models which have been pretrained on larger training datasets like Llama. The advantage of InfillingScore in detection accuracy comes from the fact that it incorporates tokens from both past and future tokens in the sequence, which allows it to determine outlier (seen) examples more accurately.
> So to summarize: Yes indeed our method will take a few more seconds, but this is worth it for contamination detection, in our opinion.

---

> ### Author Response · Authors · 2024-11-26
>
> Dear Reviewer kRF2,
>
> We hope the above clarifications and the additional experiments in the revised draft addressed your concerns about the novelty of this work. We kindly request you to discuss if there are any further comments as we remain committed to addressing any remaining points you may have during the discussion phase.
>
> Thanks.

---

> > ### Comment · Reviewer_kRF2 · 2024-11-28
> > **Reviewer Response**
> >
> > Thank you for the response and sorry for the late reply. After reading your response and others' reviews, I think my concerns are not resolved.
> > 1. Contributions are marginal: Applying Bayesian probability for output tokens and the approximation of exact infilling scores to the existing Min-k% and Min-k%++ method is hardly a contribution to me. It is simply adding two intuitive tricks to an existing framework that is shown to be not effective for general applications in the past and recent literature [1, 2, 3] (including these papers but not limited to) that the token probability-MIA methods are not performing well for LMs and LLMs. Therefore, I don't think this paper has made substantial amount of contributions.
> >
> > 2. The performance is not statistically "much better" than the baseline methods as claimed in the paper and rebuttals, where the improvement is very marginal considering the binary classification tasks. In the experiments from Sec 4.4.2, Infilling Score did not perform well for about half the cases. Considering the conclusions from [1] where most methods perform nearly random for some MIA evaluations, I don't think this paper can be applied to real downstream tasks.
> >
> > Therefore, I think my current evaluation is reasonable for this paper and will keep the current ratings.
> >
> > [1] Duan, Michael, et al. "Do membership inference attacks work on large language models?." arXiv preprint arXiv:2402.07841 (2024).\
> > [2] Carlini, Nicholas, et al. "Extracting training data from large language models." 30th USENIX Security Symposium (USENIX Security 21). 2021.\
> > [3] Carlini, Nicholas, et al. Is ami (attacks meet interpretability) robust to adversarial examples? arXiv preprint arXiv:1902.02322.

---

### Author Response · Authors · 2024-12-03

We thank all the reviewers for their effort in helping us improve our manuscript. We are glad that most reviewers have found the proposed method novel and experiments to be extensive.


We want to emphasize on the main contribution of the work again here. Using the non-causal likelihoods is a natural and intuitive method for improving the accuracy of methods like MinK%++. However, it is not trivial and  how to compute this statistic with a feasible algorithm which does not require order |V| number of LLM calls (where |V| is the vocabulary size). The innovation of our work is in introducing the ratio statistic that enables the method to incorporate non-causal token likelihoods.

We have shown empirical results in all existing datasets which are used for membership inference in the literature. Our method achieves a higher accuracy compared to existing membership inference methods. There are very few exceptions which are listed in our results. Based on reviewers' great suggestions, we have also conducted a bootstrap hypothesis test and calculated the two-sided p-value to indicate the statistical significance of our method. In addition, we have also clearly indicated the limitations of our work (in runtime) compared to existing methods, and have measured and reported the comparative runtime results.

We want to thank the reviewers again for their comments and suggestions towards improving this work.

---

### Meta-Review · Area_Chair_4qTt · 2024-12-20

**Metareview:**

The proposed Infilling Score method contributes novel technical innovations by incorporating both past and future token data for more accurate contamination detection, addressing existing shortcomings in the current methods that rely solely on past tokens. The paper provides a comprehensive experimental evaluation across multiple datasets, demonstrating an improvement over previous methods. The improvements, particularly for larger pretraining datasets and longer sequences, and the efficient computation of the proposed score are noteworthy achievements. The setup of a new benchmark dataset is another valuable resource for the research community.

While some reviewers described the contribution as incremental compared to Min-k% and Min-k%++, the authors contended that the contribution is non-trivial and explained how they developed a feasible algorithm for computing this statistic without necessitating an order of |V| language model calls (where |V| is the vocabulary size). The innovation in this work lies in the introduction of the ratio statistic, enabling the method to incorporate non-causal token likelihoods. I agree with the authors' assertion that a computationally feasible algorithm qualifies as a significant contribution.

**Additional Comments On Reviewer Discussion:**

The reviewers initially had differing opinions about the paper's contributions. Reviewer kRF2 regarded the contributions as marginal, citing that the improvements are minor and do not substantially advance the field. On the other hand, reviewers ZiHR and 7osK found the paper to be a solid contribution, emphasizing its novelty and performance gains over existing methods. After the author's rebuttal clarified points regarding statistical significance testing and practical applicability concerns, reviewer ZiHR was satisfied, while reviewer kRF2 maintained their stance. Reviewer T2Q9 provided a nuanced view, acknowledging the improvements but requesting clarity on specific experimental questions, which was addressed by the authors.

---

### Decision · Program_Chairs · 2025-01-22

Accept (Poster)